# Malaria surveillance reveals parasite relatedness, signatures of selection, and correlates of transmission across Senegal

Stephen F. Schaffner [1], Aida Badiane[2], Akanksha Khorgade [1], Medoune Ndiop [3], Jules Gomis[2], Wesley Wong[4], Yaye Die Ndiaye[2], Younouss Diedhiou[2], Julie Thwing[5], Mame Cheikh Seck[2], Angela Early[1], Mouhamad Sy[2], Awa Deme[2], Mamadou Alpha Diallo[2], Ngayo Sy[6], Aita Sene[2], Tolla Ndiaye[2], Djiby Sow[2], Baba Dieye[2], Ibrahima Mbaye Ndiaye[2], Amy Gaye[2], Aliou Ndiaye[2], Katherine E. Battle [7], Joshua L. Proctor[7], Caitlin Bever[7], Fatou Ba Fall[3], Ibrahima Diallo[3], Seynabou Gaye[3], Doudou Sene[3], Daniel L. Hartl [8], Dyann F. Wirth[1,4], Bronwyn MacInnis [1], Daouda Ndiaye[2] & Sarah K. Volkman[1,4,9] ✉

We here analyze data from the first year of an ongoing nationwide program of genetic surveillance of *Plasmodium falciparum* parasites in Senegal. The analysis is based on 1097 samples collected at health facilities during passive malaria case detection in 2019; it provides a baseline for analyzing parasite genetic metrics as they vary over time and geographic space. The study's goal was to identify genetic metrics that were informative about transmission intensity and other aspects of transmission dynamics, focusing on measures of genetic relatedness between parasites. We found the best genetic proxy for local malaria incidence to be the proportion of polygenomic infections (those with multiple genetically distinct parasites), although this relationship broke down at low incidence. The proportion of related parasites was less correlated with incidence while local genetic diversity was uninformative. The type of relatedness could discriminate local transmission patterns: two nearby areas had similarly high fractions of relatives, but one was dominated by clones and the other by outcrossed relatives. Throughout Senegal, 58% of related parasites belonged to a single network of relatives, within which parasites were enriched for shared haplotypes at known and suspected drug resistance loci and at one novel locus, reflective of ongoing selection pressure.

Despite progress over the past several decades toward malaria control and elimination, *Plasmodium falciparum* malaria remains a major global cause of human morbidity and mortality. As countries seek to improve the targeting and effectiveness of malaria interventions, they require detailed information about the prevalence of drug resistance, about local epidemiology, and about how the malaria burden is changing in response to interventions and other factors. Parasite genetic surveillance shows great promise as a source of information to support decision-making in all of these areas[1–4]. It already has a well-established role in tracking the spread of known drug resistance markers and detecting the appearance of new resistance alleles[5,6]. In low transmission settings, genetic data can determine whether new

cases arise from ongoing local transmission or from importation, and it can potentially identify sources of imported parasites[7–9].

Detecting changes in malaria burden is challenging, both because directly surveying parasite prevalence is difficult and because indirect estimates can be skewed by varying care-seeking behavior and access to health resources. Here again, genetic surveys have the potential to provide important information since parasite genetics has already been observed to reflect changes in transmission rates[10]. However, the relationship between malaria burden and genetics is complex, in part because the parasite life cycle includes both a haploid asexual stage in humans and a diploid sexual stage in the mosquito. Both self-fertilization and outcrossing can occur during the sexual cycle, with the latter requiring the presence of multiple genetically distinct parasites in the mosquito, something that occurs more frequently when transmission is higher. Thus, as malaria transmission drops, we can expect a decrease in the typical complexity of infection (COI, the number of distinct parasite genomes present in an individual) and an increasingly clonal parasite population, both of which have been observed[11]. The exact relationships between these aspects of parasite population genetics and transmission, and whether they are able to provide useful information for malaria control, are still being investigated. Even less is known about the utility of partially related parasites (close relatives that are not clones) in understanding transmission; these should become less common with declining transmission (and hence declining outcrossing) but also easier to detect in a smaller parasite population.

Here we describe an analysis of the relationship between parasite transmission dynamics and parasite genetics, based on the first year of *P. falciparum* genetic surveillance across Senegal. The project is intended as a testbed for learning how to apply parasite genetic surveys to inform malaria control strategy across the range of malaria transmission intensity seen in Senegal, which is stratified into three zones (Fig. 1a), with very low transmission in the north of the country (annual incidence <5 cases/1000/year), more moderate transmission in the middle (5–25/1000), and higher transmission in the southeast (>25/1000). Much of Senegal is approaching pre-elimination status, but persistent high transmission in the south (>500/1000 in some catchment zones) together with the importation of infections into low transmission zones threaten these gains.

The dataset consists of genetic data from 1097 *P. falciparum* samples collected in 2019 from 23 health facilities in Senegal, primarily as part of the National Malaria Control Program's (NMCP's) sentinel surveillance network. All samples were subjected to barcode genotyping based on 24 single nucleotide polymorphisms (SNPs), which were used to identify clonal parasites and polygenomic infections

(those with COI > 1). A subset of primarily monogenomic samples was chosen for more detailed analysis using whole genome sequencing (WGS) and subsequent genome-wide SNP genotyping, which gave us the ability to detect partially related parasites and identify signatures of selection. Our goal was broadly to explore ways in which this genetic data could be informative for malaria control decision-making, and in particular to investigate the relationship between parasite genetics and malaria burden.

## Results

We collected samples for genetic analysis during the malaria transmission season in Senegal (July to December 2019) from febrile individuals with uncomplicated malaria infections at 23 health facility sites that serve as national sentinel sites across the country (Fig. 1 and Table 1); these provide a broad representation of transmission intensity. For each site, we also obtained data on local malaria incidence as determined by the NMCP's routine surveillance program[12]. The final dataset used for analysis comprised 1097 samples: 310 with both barcode and genome-wide SNP data, 757 with barcode data only, and 30 with genome-wide SNP data only (their barcode data having failed our quality threshold for analysis). Samples chosen for genome-wide analysis were subjected to *P. falciparum* selective whole genome amplification and Illumina-based short-read whole genome sequencing. Monogenomic samples were analyzed at ~40,000 polymorphic genome-wide SNPs filtered to remove highly heterozygous regions and SNPs (see Methods and Supplementary Fig. 1 for details on filtering); when multiple clones were sequenced from a study site, one randomly selected sample was included in the analysis.

### Parasite relatedness varies with distance and reveals distinct patterns for two sites of similar incidence

We first examined parasite genetic diversity, assessing in particular whether allele frequencies varied significantly throughout Senegal. As a measure, we used the pairwise genetic distance between parasite genomes (i.e., the heterozygosity measured only at our trusted SNP loci). We found no evidence for systematic differences in allele frequencies and thus no evidence for parasite population structure within Senegal. Mean parasite diversity between study sites was independent of the distance between sites and was indistinguishable from diversity within sites (Fig. 2a).

Since allele frequencies provided no geographic information about parasites, we turned to parasite relatedness, which has elsewhere been shown to vary with distance and to be informative about parasite migration (e.g., ref. 13). Pairs of relatives were identified either as clones, based on barcode data, or as partial relatives, based on the

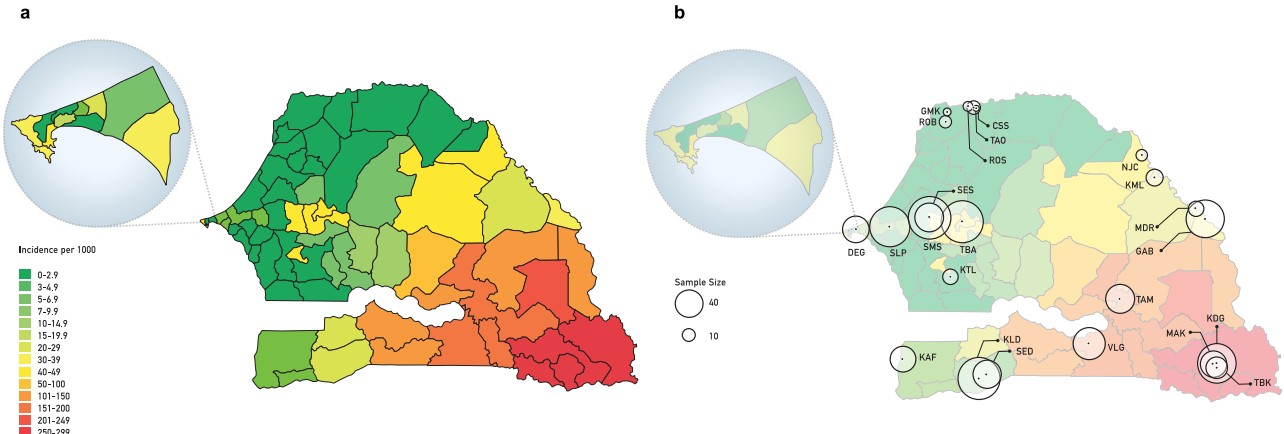

**Fig. 1 | Map of Senegal showing incidence and location of sample collection sites. a** Map of Senegal showing annual malaria incidence per 1000; **b** location of sample collection sites with relative sample size by site. Inset expands the Dakar region.

**Table 1 | Dataset and site information**

| Codes | | Location | | | WGS | Barcode | |
|---|---|---|---|---|---|---|---|
| Site | Map | Region | District | Incidence | M | M | P |
| GMK | STL | Saint Louis | Richard Toll | 0.5 | 0 | 4 | 0 |
| ROS | STL | Saint Louis | Richard Toll | 0.8 | 0 | 5 | 2 |
| KTL | KTL | Kaolack | Ndoffane | 1.5 | 6 | 9 | 10 |
| TAO | STL | Saint Louis | Richard Toll | 1.6 | 0 | 2 | 0 |
| NJC | STL | Matam | Matam | 2 | 3 | 4 | 6 |
| ROB | STL | Saint Louis | Richard Toll | 2.1 | 0 | 9 | 3 |
| CSS | STL | Saint Louis | Richard Toll | 2.1 | 0 | 10 | 12 |
| SLP | SLP | Thiès | Thiès | 2.3 | 40 | 51 | 19 |
| DEG | DEG | Dakar | Pikine | 2.4 | 14 | 26 | 11 |
| MDR | MDR | Tambacounda | Bakel | 4.4 | 5 | 13 | 9 |
| SED | SED | Sedhiou | Sedhiou | 4.7 | 18 | 23 | 11 |
| SES | DBL | Diourbel | Diourbel | 7.5 | 13 | 90 | 12 |
| KAF | KAF | Ziguinchor | Diouloulou | 13.1 | 16 | 23 | 20 |
| KML | KML | Matam | Kanel | 13.2 | 7 | 10 | 11 |
| SMS | DBL | Diourbel | Diourbel | 15.2 | 3 | 39 | 15 |
| TBA | TBA | Diourbel | Touba | 19.1 | 41 | 61 | 17 |
| GAB | GAB | Tambacounda | Bakel | 64.3 | 33 | 47 | 25 |
| VLG | VLG | Kolda | Velingara | 127.8 | 21 | 41 | 53 |
| KLD | KLD | Kolda | Kolda | 208.3 | 32 | 70 | 40 |
| KDG | KED | Kedougou | Kedougou | 414.8 | 45 | 42 | 39 |
| TAM | TAM | Tambacounda | Tambacounda | 611.2 | 13 | 40 | 35 |
| TBK | KED | Kedougou | Tomboronkoto | 937.1 | 13 | 16 | 18 |
| MAK | KED | Kedougou | Kedougou | 1076.9 | 17 | 26 | 35 |
| TOTAL | | | | | 340 | 661 | 403 |

Breakdown of samples from each site in the study. Each site has a three-letter code; in some cases, multiple sites are clustered into a single site with its own code for display on maps. Incidence: annual incidence per 1000; WGS: number of samples with whole genome sequence data; Barcode: samples with barcode genotyping data broken down by their classification as monogenomic (M) or polygenomic (P). Three sites with a single barcode sample have been omitted.

fraction of their sequenced genomes that were identical by descent (IBD), that is, the fraction that consisted of chromosome segments with essentially identical alleles throughout[14–17] (see "Methods"). If the IBD fraction exceeded a threshold (4 or 5% of the genome, depending on sequencing depth—see Methods and Supplementary Fig. 2 for details), they were classified as relatives. Related parasites were recovered in 13/23 study sites (12/15 sites with at least 25 samples) as well as between sites at all distance scales across the country (Fig. 2b). In total, 2789 sample pairs were classified as related, amounting to 4.8% of all possible pairs. Unlike allele frequencies, the probability that a pair of parasites were related to each other (which we term 'pairwise relatedness') showed marked spatial structure, decreasing from 3.9% for parasites within the same site to less than 0.1% for separations greater than 300 km (Fig. 2c). While this trend conforms to expectations, we note that there is a large variance between sites separated by similar distances.

Relatedness was particularly high throughout a low to moderate transmission region across central/western Senegal. This area is a good candidate for high human (and therefore parasite) movement between sites. It has the highest human population density within Senegal as well as the most well-developed road network. In addition, many of our study sites in this region have specific characteristics that increase connectedness with the rest of the country, whether because they are in or near the capital (DEG, SLP), a major destination for religious education or pilgrimage (KTL, TBA, DBL), or located at important road crossings (TBA, KTL). While the high degree of parasite relatedness between these sites must reflect parasite movement, it should also be

noted that all sites that have a high pairwise relatedness with other sites also display high levels of within-site relatedness (Fig. 2b; the possible exception, KTL, has too small a sample size for reliable estimation of relatedness). The high intra-site relatedness could reflect in part the fact that high within-site relatedness can make it easier to detect external relatives: descendants of imported parasites are more detectable in low transmission regions (because they make up a larger fraction of the local population) and persist for more generations (because there is less outcrossing). Thus, while we can conclude that parasite movement is common in this part of Senegal, we cannot conclude that it is rare elsewhere.

The relatives identified here are of two distinct kinds, clones, and partial relatives; as noted above, these have different relationships to transmission intensity, with partial relatedness more likely to occur where transmission is higher. We, therefore, investigated the two components separately, calculating *clonality* (the fraction of parasite pairs that are clones of each other) and *partial relatedness* (the fraction of non-clonal pairs that contain relatives) within each site (Fig. 2d). The two kinds of relatedness showed little correlation; in fact, the two cities with the highest overall relatedness had strikingly different patterns of relatedness. Touba (study site TBA, total relatedness = 0.137) had very high partial relatedness but unremarkable clonality. By contrast, Diourbel (study sites SES and SMS, total relatedness = 0.157), had clonality that was several-fold higher than in any other site and partial relatedness that was difficult to estimate because there were so few non-clonal parasites. The clonal parasites in Diourbel did not represent a single clonal expansion but instead occurred in numerous distinct clusters of varying size (Fig. 2e). Consistent with the finding of high clonality, the polygenomic fraction observed in the main study site in Diourbel (SES) is the lowest of any of our sites (0.12), implying that there is little opportunity for outcrossing to occur locally. The abundance of outcrossing evident in Touba, on the other hand, occurred despite the second-lowest polygenomic fraction observed (0.22).

**Transmission intensity is more correlated with the frequency of polygenomic infections than with relatedness**

We next addressed one of our core goals, which was to investigate how well genetic measures can be used to estimate the changing malaria burden. Previous work has shown that relatedness increases with decreasing transmission[10,11], raising the possibility that measurements of relatedness could help guide malaria control efforts. To study the relationship between relatedness and transmission, we used a proxy for transmission, the reported malaria incidence at each site; this was based on case data routinely collected by the Senegal NMCP and calculated as the ratio of case counts per year to estimated catchment populations[12]. As predicted, pairwise relatedness was negatively correlated with incidence (Fig. 3a). However, the correlation was not strong (Pearson's $r = -0.44$) and the relationship was not at all linear. Partial relatedness and clonality calculated separately were even less correlated with incidence ($r = -0.34$ and $r = -0.37$, respectively). Primarily, the observed pattern suggests that at higher incidences (with a threshold in the range of 10–50 per 1000 per year), relatedness is always low, while at lower incidences it can lie in a wide range.

Another, potentially more promising, aspect of genetics for tracking transmission[18] is COI, which has been proposed to offer both more accurate and more current information about transmission changes. The measure we investigated for this purpose was the frequency of polygenomic infections at each study site (termed the *polygenomic fraction* hereafter), based on the large number of samples for which we had barcode data. Given the small number of SNPs in the barcode, we treated COI as a dichotomous trait, with samples classified as "probable monogenomic" (<2 heterozygous SNPs) or "probable polygenomic" (≥2 heterozygous SNPs, see Methods and Supplementary Fig. 3 for details).

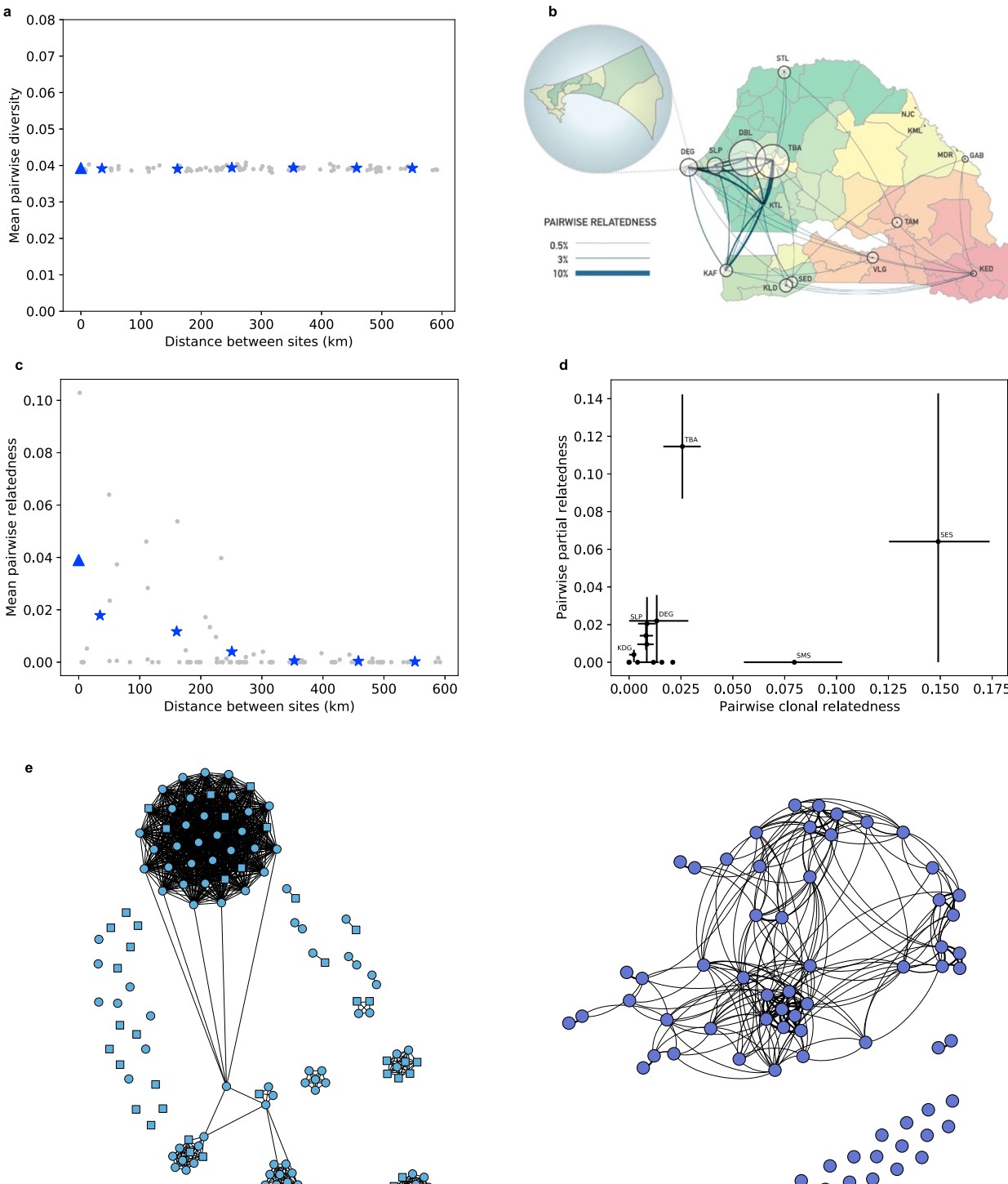

**Fig. 2 | Genetic relatedness within and between study sites. a** Mean pairwise genetic diversity measured within (triangle) and between (stars) study sites as a function of distance (kilometers, km). Gray dots give values for individual site pairs, while triangles give weighted averages in 100 km bins. **b** Pairwise relatedness within and between sites, indicated by area of circles and thickness of lines, respectively. **c** Mean pairwise relatedness within (triangle) and between (stars) study sites as a function of distance (km). **d** Partial relatedness vs. clonality for each study site (see Table 1 for site codes and sample sizes; error bars represent 68% confidence intervals, see Methods). **e** Networks of related parasites in Diourbel (left) and Touba (right). The two study sites in Diourbel are indicated by circles (SES) and squares (SMS). Line thickness is proportional to the degree of relatedness, with the thickest lines indicating clones.

Broadly, the polygenomic fraction correlated with transmission intensity across the country, decreasing from ~50% in the highest burden region in the southeast to 10–30% in the central region, but increasing again in the extremely low transmission region in the far north, where the fraction was 40% (Fig. 3b and, on a log scale,

Supplementary Fig. 4). The correlation with measured incidence was moderately strong (Pearson's $r = 0.77$). The breakdown of the correlation at very low incidence is consistent with the previously reported behavior of COI[19]. The relationship was little changed when we adjusted the raw incidence estimates for differing levels of care seeking,

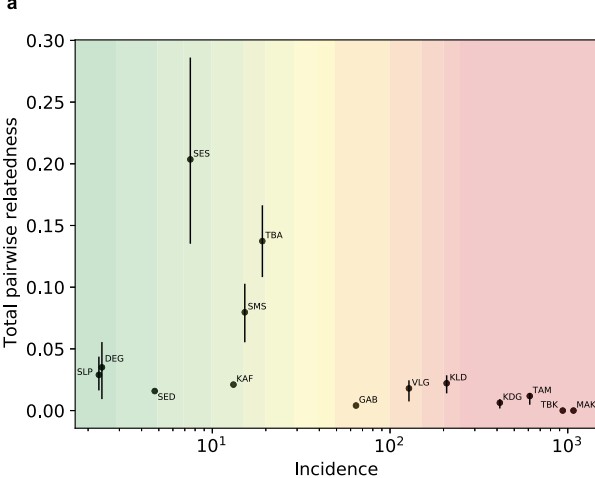

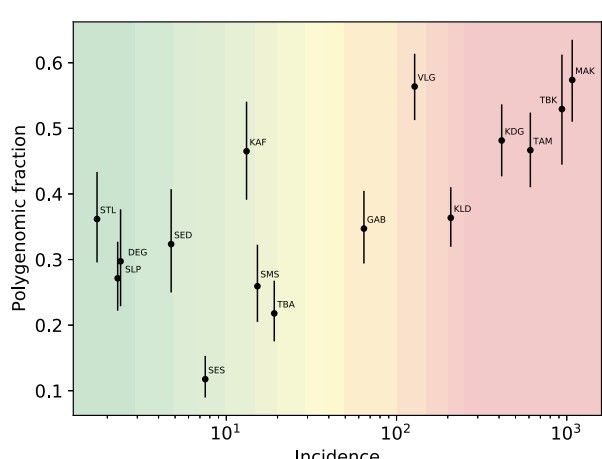

**Fig. 3 | Relationship between genetic metrics and incidence. a** Relationship between pairwise relatedness and reported incidence at study sites (see Table 1 for site codes and sample sizes). **b** Relationship between polygenomic fraction and reported site incidence. In **a** and **b**, error bars represent 68% confidence intervals arising from sampling error (see Methods for details).

testing of suspected cases, and reporting completeness at our study sites, using a previously described approach[20] (Supplementary Fig. 5); the most notable change was a seven-fold increase in the estimated incidence at the Velingara (VLG) site, making its estimated incidence more consistent with the observed polygenomic fraction.

The polygenomic fraction was more informative about incidence than relatedness, whether the latter was measured as total relatedness or separately as partial and clonal relatedness. When we regressed $\log_{10}$(incidence) on the polygenomic fraction and on relatedness, the former was the much better single predictor ($R^2_{adj} = 0.527$ for polygenomic fraction vs. 0.127 for total relatedness, see Supplementary Table 1). Combining relatedness with the polygenomic fraction provided little additional information ($R^2_{adj} = 0.550$), as did replacing relatedness with its partial and clonal components ($R^2_{adj} = 0.507$).

**Parasite relatedness reveals a network of connectedness across Senegal and loci under selection**

When we grouped sequenced parasites from throughout the country into networks of partial relatives, we found a strikingly skewed distribution of network sizes. Omitting clones, there were 110 parasites with a relative in the dataset or 32% of the total. These formed a total of 22 clusters of relatives. Twenty-one clusters contained two or three parasites and together accounted for 46 of the 110. The remaining single cluster contained the other 64 parasites (58% of partially related parasites) and included parasites from eight study sites across central and southern Senegal (Fig. 4a). This observation seems counterintuitive, since the large fraction of parasites in a single cluster suggests a small parasite effective population size, while the large number of partially related parasites requires frequent, repeated superinfection, something that does not normally occur in small populations.

One process that could contribute to this kind of clustering of relatives is ongoing positive selection, with increased relatedness driven by the sharing of genomic segments under selection. This possibility would also be consistent with the observation that parasites within the large cluster were twice as likely to have a clone in our dataset as those from the same study sites but outside the cluster— 26.7% (16/60) vs. 12.5% (14/112) ($p = 0.033$, two-tailed Fisher's exact test)—although the marked variation in clonality and partial relatedness between sites makes interpretation difficult. To address this possibility, we looked for genome segments that were more widely shared (based on our IBD analysis) between cluster parasites than between non-cluster parasites. While we found no specific segments shared among the majority of cluster parasites, we did find multiple regions of increased sharing within the cluster (Fig. 4b). The most pronounced of these contain genetic loci known or suspected to be involved in drug resistance. For example, the *P. falciparum* chloroquine resistance transporter (*pfcrt*) locus (PF3D7_0709000) on chromosome 7[21] is known to modulate drug resistance, the amino acid transporter (*aat1*) gene (PF3D7_0629500) on chromosome 6 has been implicated in drug resistance[22], and the GTP cyclohydrolase 1 (*gch1*) locus (3D7_1224000) on chromosome 12 has been implicated in antifolate resistance[23,24]. In all of these cases, enhanced sharing was evident in both cluster and non-cluster-associated parasites but was more pronounced among those in the cluster. The same was true of a region on chromosome 9 that has been reported previously to be under selection in Senegal and Gambia[25] but that has not been associated with drug resistance. A ~ 200 kb region, containing dozens of genes, on chromosome 11 showed excess sharing only among parasites in the cluster, raising the possibility that it is an early signal of an emerging sweep (see Supplementary Materials for pileup plots for all chromosomes).

## Discussion

While parasite genomic surveillance shows growing promise for informing national malaria control strategies, it is still limited by uncertainty about operationally informative metrics of genetic diversity. Here we describe a broad survey of *P. falciparum* genetic diversity across Senegal, collected through the NMCP's sentinel system, which offers an epidemiological framework for analysis of the genomic data. The data provide a country-wide baseline of circulating parasite diversity for ongoing surveillance and a test set for identifying robust analytical approaches and meaningful metrics for understanding malaria transmission.

We found no evidence of parasite genetic differentiation between regions of the country. In contrast, the distribution of related parasites, which reflects only very recent population history, showed a clear geographic structure. Given the large historical population size of *P. falciparum* in the region, the minimal population structure is not surprising. Less obvious is whether ongoing migration within the country will be enough to maintain that homogeneity despite currently small parasite population sizes across much of the country; this should become clearer with data from subsequent years.

We investigated the relationship of epidemiological incidence to statistics measuring different aspects of parasite genetics. This effort was motivated by the challenge of accurately monitoring malaria incidence[26], especially where transmission is low: direct

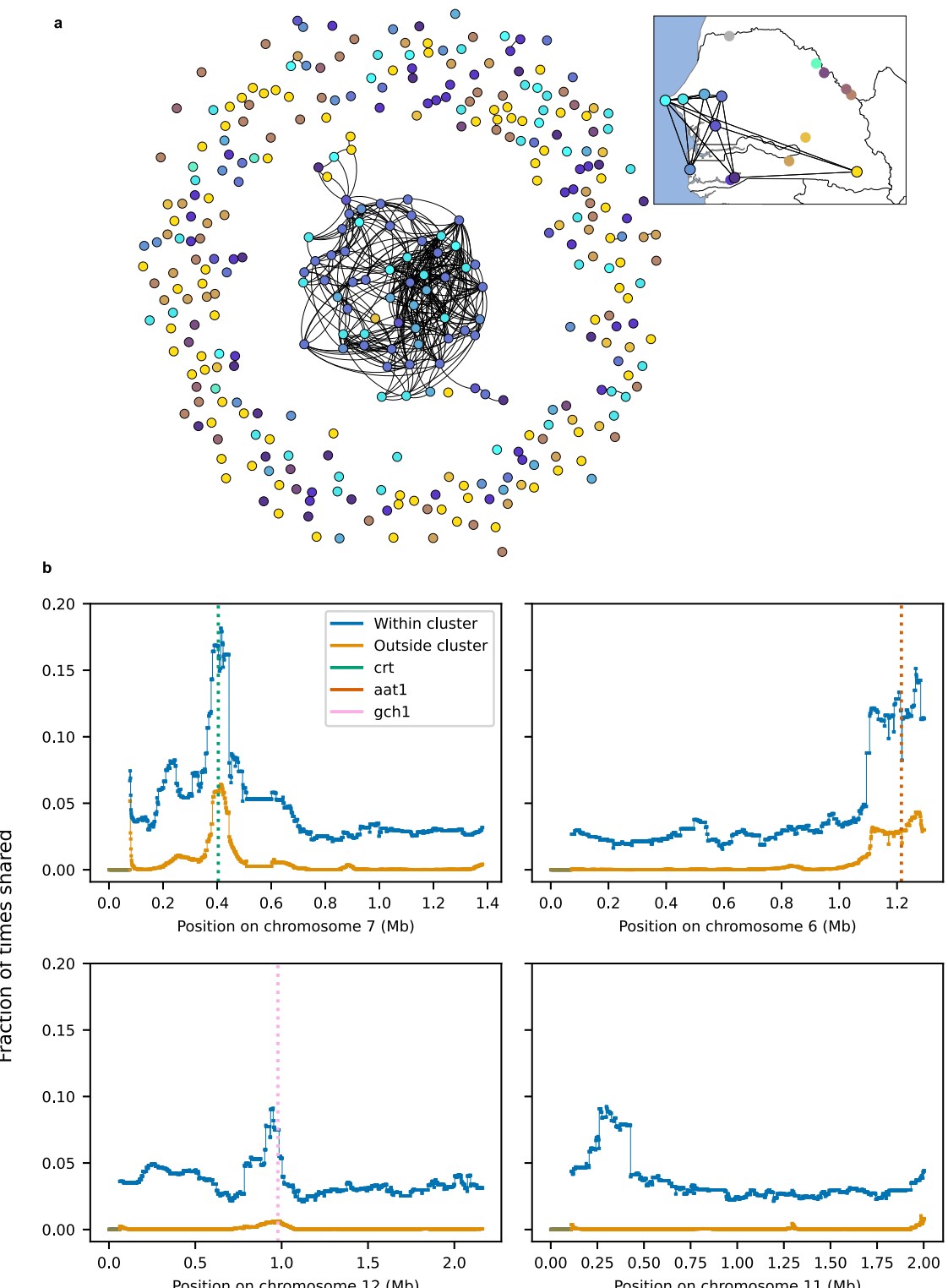

**Fig. 4 | Related parasites share genomic regions of reduced diversity consistent with a selective sweep.** Parasite relatedness between study sites. **a** Network of relatedness between all sequenced samples (clones excluded). Samples are colored by study site as indicated on the inset map. **b** The fraction of sample pairs that are IBD as a function of genomic position, split into samples that are ("within cluster") or are not ("outside cluster") part of the large network of related parasites. The four examples shown are a selective sweep around the known drug resistance locus at pfcrt (crt, green dotted line); a suspected sweep on chromosome 6 surrounding the aat1 locus (brown dotted line); a possible sweep on chromosome 12 containing the gch1 locus (pink dotted line); a possible unreported sweep on chromosome 11.

measurement of the entomological inoculation rate becomes impractical when the rate is low, while estimates based on reported cases are biased by varying care-seeking behavior and access to health resources.

Identifying genetic metrics that could serve as proxies for incidence would be of practical value for informing malaria control efforts. Unfortunately, the same challenges also hinder efforts to assess the relationship between genetics and incidence, since we generally do not

have firm estimates of the true incidence to act as a truth set. The size of the uncertainty is illustrated by one of our study sites, that in Velingara, for which correcting for biases in the reported incidence increased the estimated incidence by a factor of seven. Thus, a good understanding of the relationship between incidence and parasite genetics is likely to require multiple studies under different conditions as well as detailed modeling. For the purpose of this project, however, a complete understanding of that relationship is not critical; our focus will be on changes in genetic metrics over time since these are likely to be informative about changes in incidence even without perfect knowledge of the true incidence.

The statistics we compared to incidence were the frequency of relatedness (partial or complete) between parasites and the frequency of complex (polygenomic) infections in the population. We drew two main conclusions from the comparison. First, while both measures were correlated with local incidence, the polygenomic fraction was a much better predictor than relatedness, confirming modeling studies that identified COI-based statistics as superior to relatedness- or diversity-based ones[18,27]. In particular, COI-based measures are predicted to respond more rapidly to changes in transmission[18], making them good candidates for monitoring the effect of interventions in moderate to high transmission settings. Notably, the polygenomic fraction was the better predictor despite the limitations of low-resolution genetic data from the 24-SNP barcode, which did not allow us to estimate the number of genomes in polygenomic infections. This illustrates the continuing value of low-resolution genetic data in capturing information about transmission and raises the hope that data with higher resolution for COI will be even more informative.

We also found that, while the correlation between either genetic measure and local incidence holds true for moderate and higher transmission regions, it breaks down where transmission is low (roughly, when incidence <50/1000/year). In our low-incidence sites, genetic measures suggest a much higher incidence than is actually observed; similar behavior has been previously reported specifically for COI[19]. In the far north of Senegal, incidence is so low (~1/1000) that the same individual should rarely be infected twice simultaneously, and yet 40% of infections in the region are polygenomic. Based on the known epidemiology and a previous study of parasite genomics of this region, it seems likely that many parasites there represent recent importation from areas with higher burden[7,27] and that polygenomic infections are either acquired elsewhere or are the result of local co-transmission of multiple parasite strains from a single imported polygenomic infection. It is thus not surprising that local parasite genetics in this part of Senegal reflect the characteristics of the source region more than the local transmission rate[28]. Similarly, relatedness should be low where the bulk of infections represent recent importation from more diverse regions, and where it does appear (typically in the form of clones) it signals ongoing local transmission. A better understanding of the transition of genetics into an importation-dominated regime will likely require both more in-depth study of particular regions and detailed modeling.

Despite its limitations as a predictor of transmission intensity, parasite relatedness proved to be illuminating about the heterogeneity of malaria transmission in Senegal. An unexpected finding was that two geographically close (~50 km apart) cities, Touba and Diourbel, with similarly high overall parasite relatedness, differ markedly in the patterns of that relatedness, with partially related parasites common in Touba while clones dominate in Diourbel. Touba does have a higher reported incidence than the main study site in Diourbel (15/1000 vs. 8/1000), but the different transmission dynamics in the two cities may also reflect their differences as settings for malaria transmission. Touba, the second most populous city in Senegal, is located at the crossing of major roadways and is the site of the annual Grand Magal pilgrimage, during which more than 4 million individuals travel to the area[29] and mix with more than 1 million local inhabitants; in 2019, this took place on October 16–17, roughly in the middle of the sample collection period. Diourbel, on the other hand, is a smaller urban setting (estimated population around 100,000) that contains numerous large (500–1000 individuals) religious boarding schools called daaras, where up to several hundred school-aged boys, many from elsewhere in Senegal, live communally. Previous work has shown evidence of distinct, genetically identical parasites among infections from individual daaras elsewhere in Senegal, consistent with local and focal transmission[9]. The findings here have helped lead Senegal's NMCP to start developing new strategies to address the infection risks associated with large population movements like the Grand Magal and have motivated a PMI-funded operational research study that will evaluate new interventions specifically targeting these at-risk students and increased community case management targeting these schools.

Subsequent surveillance data should provide more fine-grained temporal data about transmission in Touba and the role that the Grand Magal pilgrimage plays in local transmission. This pilgrimage may serve as a natural experiment for studying the effect of malaria importation at different points in the transmission season; it draws millions of people from throughout Senegal and the broader region to a single city for a brief period, and that period shifts by ~11 days each year. Further data will also show whether the large number of clones and clonal clusters in Diourbel is a persistent feature and, if so, whether the same clones persist across the dry season or are replenished by new imported infections each year. Modeling and epidemiological investigation could both provide insights into the cause of the pattern in Diourbel, in particular investigating the possibility that it is driven by highly localized hot spots of transmission. Such hot spots could then be addressed with targeted, vector-based interventions. More broadly, these observations illustrate both the potential for genetic surveillance to provide actionable information about local transmission patterns and the need to better understand the implications of specific patterns.

Finally, another striking observation from the country-wide dataset was that clusters of partially related parasites had a highly skewed size distribution, consisting of 21 small clusters (2–3 parasites apiece) and a single large cluster connecting multiple sites and containing 64 parasites. Since this kind of pattern could signal the rapid spread of one or more positively selected alleles, we looked for genomic loci where IBD sharing was unusually common (a signature of positive selection[15]), and in particular, was more common among related parasites. This revealed multiple signals of selection. Most of these were at loci known (chromosomes 4, 6, 7, and 8) or suspected (chromosome 12) to be under selection for drug resistance, while one was at a previously identified locus on chromosome 9 that is likely not associated with drug resistance. The excess IBD sharing on chromosome 6 was consistent with a recent report[30] from Gambia of the rapid rise of nonsynonymous mutation S258L in the gene *aat1*: this allele was present in 100% of common shared haplotypes in our data, while its frequency was 79% in the dataset as a whole. No single selected allele was responsible for all relatedness in the network of relatives, but several loci had markedly higher sharing within the network, and a signal at one candidate locus on chromosome 11 was seen only among samples in the network.

These enhanced signals for selections among relatives could result either from enrichment for currently selected alleles among relatives or from higher frequency of selected alleles in the central part of Senegal where most of the related parasites were found; in either case; ongoing selection would be implicated. Since positive selection in *P. falciparum* is often driven by drug resistance alleles, we view these loci as targets for continued monitoring. This is especially the case for the chromosome 11 locus. The signal there, which extends over ~200 kb, has not appeared in previous data from Senegal and we have not identified an obvious candidate gene among the ~50 covered by it. Data from subsequent surveys will clarify whether the signal

represents a statistical fluctuation or the first sign of a new selective sweep. More broadly, this finding suggests that searching for signals of selection among related parasites could be a valuable tool for the surveillance of existing and new drug-resistance loci, an approach that we are actively investigating.

## Methods

### Ethics statement

Samples were obtained from febrile patients who presented at health facilities for care. Informed consent was obtained from all study participants. The study protocol was authorized by the Ministry of Health and Social Action in Senegal (SEN 19/49) and approved by the Institutional Review Board of the Harvard T.H. Chan School of Public Health (IRB protocol 16330). The CDC Human Research Protection Office reviewed the protocol and determined the CDC to be non-engaged.

### Inclusion statement

This work represents a collaboration between Senegal research (CIGASS at UCAD) and the National Malaria Control Program (NMCP), with Harvard T.H. Chan School of Public Health and The Broad Institute, and other partners. All authors, which include local (Senegalese) researchers, participated in the work represented here from study design and implementation through data generation and analysis. All data are owned by the country of Senegal under the Ministry of Health and Social Action. The capacity to carry out the genotyping and sequencing data generation is held at CIGASS within the University Cheikh Anta Diop in Senegal and the data are analyzed jointly across the collaboration.

### Sample collection

Blood samples were collected from clinics and health posts across Senegal from uncomplicated malaria infections detected by microscopy (clinics) or rapid diagnostic testing (RDT, sentinel sites). These samples are biologically unique and were fully utilized for the data generation described. Collection sites were across the range of transmission intensity in Senegal, from very low reported annual incidence (<1 per 1000) in the northwest to high reported annual incidence (>500 per 1000) in the southeast, including three main clinic sites, located in Thies, Diourbel, and Kedougou, that were augmented by health posts as part of the sentinel survey system of the National Malaria Control Program, with additional sites located in areas of interest. Blood was collected on Whatman 903 ProteinSaver filter paper material.

Incidence per thousand individuals in 2019 was calculated for each health facility using data provided by the Senegal NMCP[12]. We estimated the incidence from nearby sites for two sites (SLP and CSS) for which we did not have incidence data. The ROB site was used for CSS in Richard Toll, and a combination of incidence data from Medina Fall 1 and 2 was used as a proxy for the SLAP clinic in Thies.

### DNA extraction, genotyping, sequencing

**Sample workflow.** Nucleic acid material was extracted from 1353 blood spots collected on filter paper, with 1066 of these samples subjected to pre-amplification, before molecular barcode genotyping of all samples. A total of 1034 of the 1353 samples passed initial barcode genotyping, with 636 samples classified as monogenomic infections. Of the 1353 samples, 648 were subjected to whole genome sequencing (WGS).

**DNA extraction.** Nucleic acid material was extracted from dried blood spots collected on filter paper obtained from all microscopy- and/or RDT-positive individuals presenting at clinics or health posts with fever according to routine Senegal protocols. Briefly, dried blood spots collected on ProteinSaver cards were extracted for genomic DNA from 2 to 3, 6-mm punches using the manufacturer protocol from the Promega Maxwell DNA IQ Casework Sample kit (Promega AS1210, Promega Corp., Madison, WI).

**Genotyping.** Extracted nucleic acid material was subjected to pre-amplification[31] and molecular barcode analysis[32]. The genotypes were called by their base designation (A, T, G, or C) with missing alleles identified by "X" and working alleles where both the major and minor alleles were designated by "N". Only samples missing zero or one of the twenty-four assays were used for analysis, and monogenomic infections were called if there were zero or one of these assays called as "N".

**Sequencing.** Genotyped samples with barcode calls of up to four missing positions and no more than two "N" calls were then used for WGS. Even though the criterion for monogenomic infections was no more than one "N" call, we included samples with two "N" calls as well to increase the likelihood of including all samples that could be analyzed for relatedness (described below). Samples were subjected to sWGA[33] involving multiple displacement amplification using Phi29 polymerase followed by magnetic bead cleanup and quantification. The resultant material was subjected to fragmentation and NEBNext Ultra II FS DNA library construction according to manufacturer instructions (New England Biolabs, Beverly, MA) and then sequenced using Illumina Hi-Seq X machines.

### Variant calling

Variant calling was performed in accordance with the best practices established as a part of the Pf3K project using GATK3.5.0 and *Plasmodium falciparum* 3D7v.3 reference assembly for read alignment with bwa-mem[34]. Picard toolkit was used to mark and remove duplicates and to assess quality control metrics. For genome sequence data, individual genotypes were discarded if the supporting read was <5 reads. The downstream analysis was limited to a set of preferred SNP sites all located within the callable loci of *P. falciparum*[35].

### Selection of preferred SNP sites based on Pf3k data

To avoid artifacts caused by the high AT content and extensive low-complexity regions of the *P. falciparum* genome, we restricted our sequence analysis to a highly filtered set of 149,582 single nucleotide polymorphism (SNP) sites, where the filters (based on the global Pf3k dataset[34]) removed sites and regions with anomalously high heterozygosity.

To develop these filters, we downloaded the complete set of VCF files for Pf3k release 5[34]. Polygenomic samples were identified with DEploid[36], and removed, along with duplicate samples from Malawi and a small number of samples from Nigeria (since there were too few of the latter for within-country analyses also performed with this dataset), leaving a total of 1328 samples.

Genotype calls for sites identified in the Pf3k VCF files as single nucleotide variants were extracted without restriction to biallelic sites, provided there were at least five copies of non-major alleles across the complete set of monogenomic samples. Heterozygous calls were tabulated for each site and were used to construct a series of filters to remove regions and sites with elevated rates of heterozygosity (Supplementary Fig. 1). The filters were as follows: (1) Sites were excluded unless they were in 'core' chromosome regions, as defined in Miles et al.[35]. (2) Remaining regions were divided into non-overlapping 2 kb windows, which were filtered on the basis of mean heterozygosity in the window, with a threshold at 0.03 (Supplementary Fig. 1c). (3) SNPs within 7 base pairs of an indel were excluded (Supplementary Fig. 1d). (4) Individual SNPs with heterozygosity ≥0.04 were excluded. To the surviving set of SNPs, we added the two sites from our 24-SNP barcode that were not already included, yielding 149,582 sites used in this study. Of these SNPS, approximately 40,000 proved to be polymorphic (minor allele frequency >0.01) in our Senegal dataset.

## Sample selection and filtering

Barcodes were excluded from further analysis if they had more than one missing assay. For the sequence data, all sites with a sequencing read depth <5 in passing samples were masked out. Sequenced samples with a high rate of heterozygous calls (Supplementary Fig. 6) in the preferred SNP set were classified as possibly polygenomic and excluded from further analysis, with the threshold set at a heterozygous fraction of 0.0024. Sequenced samples for which less than 25% of sites had at least 5x sequencing depth were likewise excluded, leaving 340 monogenomic samples in the dataset for analysis. Site-based analyses were restricted to the 14 sites with at least 15 sequences each.

## Calculation of clonality and complexity of infection

Clonal pairs of parasites were identified from barcode data, with clones defined as pairs with zero mismatches between their barcodes and all other pairs defined as non-clones. See Supplementary Fig. 7 for a comparison with sequence data where sequence and barcode data overlapped.

Polygenomic samples were also identified from barcode data, as those having >1 heterozygous genotype; since barcode genotypes were used to select monogenomic samples for sequencing, the barcode data was the primary source of information about COI. This classification was corrected with sequence data when available, based on the threshold on the heterozygous fraction given above. Based on a comparison between barcode and sequence data for the same samples (Supplementary Fig. 6), we estimate that the classification of a sample as monogenomic was correct 87% of the time when based on its barcode genotype alone and 91% of the time after applying overlapping sequencing information, while 97–98% of probable polygenomic calls were correct.

## Calculation of relatedness

Clonality was calculated as the fraction of barcoded sample pairs that had zero genotype differences between them. 97% of samples identified as clones by barcode were confirmed by whole genome data in the 107 samples for which the comparison could be made.

IBD for estimating pairwise relatedness was calculated with hmmIBD[37], version 2.0.4, with options -m 20 -n 40 (maximum of 40 generations and 20 fit iterations). To study shared IBD segments in possible selective sweeps, a modified version of hmmIBD was used with the same parameters. In this version, the fraction of sites that are IBD, which is normally a free parameter in the model, was kept fixed at 50% while identifying IBD segments. This was done to remove a bias in the hidden Markov model, which is more likely to identify segments as IBD when the overall relatedness of the two samples is high. hmmIBD was modified by deleting line 949 of hmmIBD.c, version v2.0.4 ('pi[0] = count_ibd_fb / (count_ibd_fb + count_dbd_fb);'). Pileup of IBD segments across sample pairs was then calculated in 1 kb windows; a sample pair was considered IBD within that window if an IBD segment overlapped the window at all.

## IBD threshold

To identify a minimum IBD fraction that would reliably signal close relatedness between parasites, we generated IBD distributions for two sets of sample pairs, one enriched for related pairs, based on time and location of sampling, and the other similarly depleted. For this purpose, we used an expanded dataset that included sequence data from Thiès (SLP) from prior years. The enriched sample set contained pairs of parasites where each parasite was from 2019 and from one of three sites with a high degree of relatedness: SLP, SES, and TBA. For the depleted set, the parasites in each pair had to have been sampled at least ten years apart and one of them had to come from a site other than the three listed above. We compared the distribution of IBD fractions between the two sample sets, making the comparison

separately for pairs with high sequencing coverage (>50% of SNPs with at least 5x coverage) and for those with at least one parasite with lower coverage. Based on the comparison between the distributions (Supplementary Fig. 3), we defined all pairs with high coverage and with an IBD fraction >4% to be related, while for pairs with lower coverage, the threshold was set at 5%. Using these thresholds, depleted pairs were 0.6% as likely to be labeled 'related' as enriched pairs.

Total pairwise relatedness was calculated as (clonal relatedness) + (partial relatedness) * (1 − clonal relatedness), where clonal relatedness is the fraction of all pairs that are clones and partial relatedness is the fraction of non-clone pairs that are partially related. When calculating diversity and relatedness within sites and as a function of binned distance (Fig. 2b, c), sites were weighted by the number of sequenced samples (within site) or the geometric mean of the number of sequenced samples from the two sites (between sites).

## Estimating sampling error for relatedness measures

We empirically evaluated methods for calculating confidence intervals for pairwise partial relatedness within study sites. To do so, we took the expanded dataset mentioned above (this dataset plus earlier sequence data from Thiès), restricting sites to TBA, SLP, SES, and SMS in order to approximate the relatedness seen within one site, and treated the dataset as a population with measured relatedness. From it, we repeatedly drew sample sets of specified size to simulate the sample set for one of our sites and evaluated how often a particular method for calculating confidence intervals contained the true value. A standard bootstrap approach was anticonservative and biased, the latter because sampling with replacement introduces spurious relatedness. The method we identified as the most accurate was repeated downsampling of the test set without replacement, specifically, downsampling to 55% of the initial sample. Performing a similar test with the full set of barcodes, we identified 60% as the appropriate downsampled size for estimating CIs for pairwise clonality. When determining the final confidence intervals, downsampling was repeated 1000 times for each CI. Confidence intervals for relatedness between sites were determined by simple bootstrapping.

Network diagrams of related parasites were created with the software package Gephi, version 0.9.2.

## Predictors of incidence

Ordinary least squares regression was performed with the Python package statsmodel.api. The incidence was log-transformed because of the wide range in incidence values, because fold changes at all incidence levels are relevant to malaria control efforts, and because of the highly nonlinear relationship between incidence and genetic measures. The goodness of fit was assessed by $R^2_{adj}$, which corrects for the number of explanatory variables in the regression.

## Reporting summary

Further information on research design is available in the Nature Portfolio Reporting Summary linked to this article.

## Data availability

Barcode data are provided as a supplementary file. Sequence data have been uploaded to the NCBI Sequence Read Archive under BioProject PRJNA972644, with the following accession IDs: SAMN37545094 through SAMN37545096, SAMN37606032 through SAMN37606764, SAMN37663025 through SAMN37663291. The only metadata used was the identity of the collection site for samples, which is given for each sample in a file in the code Github repository (see below).

## Code availability

Relatedness calculations were done with hmmIBD, which is available from https://github.com/glipsnort/hmmIBD. Key analysis scripts for extracting heterozygosity and genotype data from a VCF file,

categorizing samples, formatting for hmmIBD, and calculating relatedness within and between sites, are available from https://github.com/glipsnort/Senegal_2019_malaria_analysis, along with intermediate files of barcode data.

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

## Acknowledgements

We would like to express gratitude to the people of Senegal, and to all the healthcare workers at the sentinel sites and clinics including medical directors, doctors, nurses, and clinic staff for their participation and support in the collection of samples required for this analysis. We thank the Senegal National Malaria Control Program in the Ministry of Health and Sanitation for their ongoing support and collaboration for these studies. We thank Kairon Shao for their help in generating the genomic data. Funding for this work was provided by the Bill & Melinda Gates Foundation (OPP1156051) to D.F.W., and the National Institutes of Health (5R21AI141843-02) to S.K.V. The findings and conclusions in this paper are those of the authors and do not necessarily represent the official position of the U.S. Centers for Disease Control and Prevention.

## Author contributions

Substantial contributions to the conception and design of the work were made by S.F.S., W.W., D.L.H., D.F.W., B.M., D.N., S.K.V. Substantial

contributions to the acquisition of data were made by A.B., J.G., Y.D., M.A.D., N.S., M.N., M.C.S., F.B.F., I.D., S.G., D. Sene, Y.D.N., M.S., A.D., A.S., T.N., D. Sow, B.D., I.M.N., A.G., A.N., S.K.V. Substantial contributions to the analysis of the data were made by S.F.S., J.T., W.W., A.K., A.E., S.K.V. Substantial contributions to the interpretation of the data were made by S.F.S., W.W., K.E.B., J.L.P., C.B., D.L.H., D.F.W., B.M., D.N., S.K.V. and S.F.S., W.W., D.L.H., D.F.W., B.M., D.N., S.K.V. drafted or substantially revised the manuscript.

## Competing interests

The authors declare no competing interests.

## Additional information

[1]Infectious Disease and Microbiome Program, The Broad Institute, Cambridge, MA, USA. [2]Centre International de recherche, de Formation en Genomique Appliquee et de Surveillance Sanitaire (CIGASS), Dakar, Senegal. [3]Programme National de Lutte Contre le Paludisme (PNLP), Dakar, Senegal. [4]Department of Immunology and Infectious Diseases, Harvard T. H. Chan School of Public Health, Boston, MA, USA. [5]Centers for Disease Control and Prevention, Atlanta, GA, USA. [6]Section de Lutte Anti-Parasitaire (SLAP) Clinic, Thies, Senegal. [7]Institute for Disease Modeling in Global Health, Bill and Melinda Gates Foundation, Seattle, WA, USA. [8]Department of Organismic and Evolutionary Biology, Harvard University, Cambridge, MA, USA. [9]College of Natural, Behavioral, and Health Sciences, Simmons University, Boston, MA, USA. ✉e-mail: svolkman@hsph.harvard.edu

