## [Peer Review File · Nature Communications]

Malaria surveillance reveals parasite relatedness, signatures of selection, and correlates of transmission across SenegalREVIEWER COMMENTS

Reviewer #1 (Remarks to the Author):

Malaria surveillance reveals parasite relatedness, signatures of selection, and correlates of transmission across Senegal

In the submitted manuscript Schaffner et al describe baseline genetic surveillance of malaria infections from across Senegal. They sample over 1,000 infections using both high (sWGA genome sequencing) and low (SNP barcode) density methods. The findings of the paper are comprehensive, and thorough. The major results are a demonstration that (1) the proportion of polygenomic infections is informative of the transmission intensity, (2) the proportion of related parasites, but not genetic diversity, is correlated with incidence, (3) local transmission can be discriminated by relatedness, (4) ongoing selection pressures at drug resistance loci have shaped patterns of relatedness. Overall, the paper provides excellent data, though stops short of achieving its goals. The framing of the work is as a baseline, and an evaluation of methods which inform incidence, and migration of parasites. What is the "actionable information for malaria control" aimed for in the paper. Why is year one of the surveillance particularly relevant to report? The data were gathered in 2019, while data for subsequent years may not yet be available there is certainly knowledge on changes in intervention programs which have been implemented in the years since.

The finding that IBD based estimates outperform allele frequency-based estimates was previously shown by Taylor, Schaffner et al PLoS Genetics, 2017. The data here provide a robust addition to those findings, though it is not clear they extend them. Figure 2B is underdiscussed, and would appear to offer some reconciliation between relatedness and geographical distance over shorter distances. KTL in particular appears to have high relatedness to many other sites – is there something unique about this location and human movement which would explain this? There is a lengthy discussion about Touba and Diourbal and human movement – while compelling and interesting as an explanation on the specific differences between sites this feels anecdotal without discussing movement around the country more broadly.

The signatures of selection in Figure 4 are interesting. There is a novel sweep around chromosome 11, though it is not clear what the target may be here. Was this detectable only here, or also in prior samples collected historically from across Senegal? Is this thought to be driven by drug resistance, or some other factor? Are there known, or suggestive, drug resistance (or otherwise interesting) genes within the locus? The description of the role AAT1 plays in chloroquine resistance has been bolstered recently by additional population genetic data, this should likely be cited here as it pertains to known selective sweeps in West Africa.

In the discussion, it states there is no population structure in Senegal. I disagree with this – rather I suggest there is no population structure detected by shifts in allele frequency, here the authors show there is considerable local relatedness which subdivide the country.

Reviewer #2 (Remarks to the Author):

This a large and excellently conducted study of genetic diversity and relatedness and its association with location and transmission intensity in a national collected in a single year / transmission in Senegal. Genomic methods and analyses are very robust and well suited for the purpose.

I only have a few relatively minor queries regarding the interpretation of the data - most of these questions relate to the analyses of the relationship between transmission intensity (here represented by incidence of clinical cases) and genetic measures.

The author correctly point out that clinical incidence is not a gold standard measure of transmission intensity but is by itself subject to many biases. However, while it is true that EIRs are much more difficult to measure at low transmission intensities, I would be reluctant to support that view that that clinical case estimate - which were used in these analyses - are becoming more

biased at low transmission settings. For many (if not most) areas of the world the opposite is true, i.e. case counts become a more reliable (surrogate) measure of true malaria transmission with decreasing transmission levels, both for health systems (easier to cover, treat and count all cases as case number drops) and epidemiological / biological reasons (e.g. acquired immunity is becoming less strong and thus case more likely to present with symptoms, increasing the proportion of infections that are detected by passive surveillance systems). To me it therefore not obvious both that low transmission setting do more urgently require alternative, genetic measure of transmission intensity nor that larger biases in incidence estimate can explain why genetic measure work less well in the lowest transmission areas.

For the relationships between incidence and measure of genetic diversity, the log-transformed plots in the supplement are much more informative than to raw data plots. The authors kind or acknowledge that by included more results based on log-transformed incidence rates in the result section that those based on raw data. I suggest including the log-transformed results into the main paper and the raw data into the supplement.

The authors explain higher that expect genetic diversity and relatedness of parasites at the lowest transmission sites with increased importation of cases resulting in measure representing more the nature of transmission at the sites of the original transmission rather than locating of sampling. This is certainly an important factor. However, other factors could also contribute: in particular spatial and/or demographic (micro) heterogeneity of transmission risk. At low transmission levels, areas of relatively high transmission may surrounded by relatively large areas of little or no transmission (could also be the case in large urban areas where local transmission maybe absent from all a few areas and thus and thus only a small part of the population is at actual risk of local transmission) and the parasite diversity indices will represent mainly this subpopulation.

On the other end of the transmission range, there is some indication that diversity indices how seem to get saturated and even higher transmission do not seem to lead to further increases

Reviewer #3 (Remarks to the Author):

This manuscript reports preliminary results of genetics of *Plasmodium falciparum* using long-established barcoding and SNP analysis protocols to assess transmission "dynamics" in different parts of Senegal in one year (2019). The authors note that there is an ongoing nationwide program as such with a goal of providing data to guide Ministry of Health level malaria control strategies. The results essentially are consistent with the past 20 years of similar research, which is primarily that complex infections (i.e. more genomes) represent more intensive transmission. This is neither new, nor definitive given the very brief time scale of the study. It is unclear how health policy can be based on such limited data. What would have been useful is using genomic information to predict efficacy of malaria vaccines, which is not addressable using neutral markers.

Response to Reviewer Comments:

We thank the reviewers for positive comments and suggestions to improve the manuscript. Detailed responses to these comments are found below.

Reviewer #1:

1. General Comment: *"...stops short of achieving its goals...what is the 'actionable information for malaria control' aimed for in the paper...why is one year of surveillance...relevant to report..."*

Perhaps we should have emphasized these more. The stated goal of the work was to: *"broadly to explore ways in which this genetic data could be informative for malaria control decision making"* [91-92]. This baseline dataset provides national data that the Senegal National Malaria Control Program (NMCP) has used *"to start developing new strategies to address the infection risks associated with large population movement like the Grand Magal and have motivated a PMI-funded operational research study that will evaluate new interventions specifically targeting these at-risk students and increased community case management targeting these schools."* [412-416]. Comparisons with this baseline data will inform changes in allele frequencies, signatures of selection, and parasite relatedness dynamics as interventions are deployed or withdrawn by the NMCP.

2. Figure 2B: *"Figure 2B is underdiscussed...is there something unique about (KTL) location..." what about "movement around the country more broadly"*.

We appreciate the suggestion. In response to the reviewer's comment, we have expanded on our reporting of results in this section, which now includes the following:

Relatedness was particularly high throughout a low to moderate transmission region across central/western Senegal. This area is a good candidate for high human (and therefore parasite) movement between sites. It has the highest human population density within Senegal as well as the most well-developed road network. In addition, many of our study sites in this region have specific that increase connectedness with the rest of the country, whether because they are in or near the capital (DEG, SLP), a major destination for religious education or pilgrimage (KTL, TBA, DBL), or located at important road crossings (TBA, KTL). While the high degree of parasite relatedness between these sites must reflect parasite movement, it should also be noted that all sites that have a high pairwise relatedness with other sites also display high levels of within-site relatedness (Fig. 2b; the possible exception, KTL, has too small a sample size for reliable estimation of relatedness). The high intra-site relatedness could reflect in part the fact that high within-site relatedness can make it easier to detect external relatives: descendants of imported parasites are more detectable in low transmission regions (because they make up a larger fraction of the local population) and persist for more generations (because there is less outcrossing). Thus, while we can conclude that parasite movement is common in this part of Senegal, we cannot conclude that it is rare elsewhere. [173 - 188]

Since we have to date been unable to obtain independent information about human movement patterns in the country, we confine ourselves to pointing out plausible contexts for high connectivity for certain study sites.

3. Figure 4: *"...novel sweep around chromosome 11...are there known, or suggestive...genes within the locus?"*

Good question. We have not been able to identify any obvious candidate genes within the locus. In addressing this locus, we are trying to strike a balance between raising the possibility of an early selective sweep and

remaining cautious because the signal could well be a fluctuation. With that goal, we have now added the following text to the Discussion:

This is especially the case for the chromosome 11 locus. The signal there, which extends over ~200 kb, has not appeared in previous data from Senegal and we have not identified an obvious candidate gene among the ~50 covered by it. Data from subsequent surveys will clarify whether the signal represents a statistical fluctuation or the first sign of a new selective sweep. [459 - 463]

4. AAT1: Add citation of recent publication that *"...pertains to known selective sweeps in West Africa"*.

We thank the reviewer for pointing out this study, which was published just after our manuscript was written. We now cite the study [Reference 30] and include the following text:

The excess IBD sharing on chromosome 6 was consistent with a recent report from Gambia of the rapid rise of nonsynonymous mutation S258L in the gene *aat1*: this allele was present in 100% of common shared haplotypes in our data, while its frequency was 79% in the dataset as a whole. [447 - 450]

5. Population Structure: *"...it states that there is no population structure in Senegal. I disagree with this..."*.

Our apologies for lack of clarity: we were using 'population structure' as shorthand for the presence of allele frequency differences between subpopulations. We now describe it as 'genetic differentiation' [341] instead.

Reviewer #2:

1. Case Estimate Bias At Low Transmission: *"...reluctant to support...view that...clinical case estimate...more biased at low transmission settings."*

We appreciate the reviewer's comment regarding bias in clinical case estimates at low transmission settings but argue that clinicians are less likely to suspect malaria where malaria is uncommon, thus less likely to test for malaria when fever is evident. Evidence from the DHS supports this conclusion, where < 1/3 of children younger than 5 years of age with fever seeking care in Senegal receive a malaria diagnostic test, compared to ~2/3 of children < 5 years in higher malaria transmission countries with similarly performing public health systems. Even within Senegal, the proportion of fevers that received a malaria diagnostic test was 5.3% in the North, where transmission is low, compared to 19.7% in the South, where transmission is higher. Finally, a recent malaria outbreak in the pre-elimination area of Matam, Senegal (Sy, PMID: 33608006) was missed by providers due to the judgment that the illness was not likely due to malaria despite the national policy of testing all fevers for malaria. Thus, perceptions by clinicians about malaria risk can impact testing and ultimately clinical case estimates.

2. Figure 3: *"include...log-transformed results in main paper and the raw data (in) supplement"*.

We're grateful for the suggestion. We have accepted this suggestion and moved log plots into the main text and made the linear plots Supplemental.

3. Genetic Metrics in Low Transmission Settings: *"...other factors could contribute..."*

We acknowledge the reviewer's comment regarding genetic metrics and transmission intensity, and that other factors contribute to the heterogeneity of transmission risk. Indeed, we provide evidence that genetic metrics can reveal just this kind of heterogeneity even at similar levels of transmission, as in Diourbel and Touba that

have strikingly different patterns of clonality or partially related infections despite similar incidence. Deviations from incidence predictions, such as observed in low transmission settings, may be related to other factors like imported infections, which are increased in some low transmission areas in Senegal due to work related migration (see Daniels, 2020; Liu, 2020; Sy, 2022). Revealing gene flow in populations can help identify sources of infection and patterns of transmission that inform intervention use. To clarify this point we have included the following text in the discussion.

For the purpose of this project, however, a complete understanding of that relationship is not critical; our focus will be on changes in genetic metrics over time, since these are likely to be informative about changes in incidence even without perfect knowledge of the true incidence. [361 - 364]

4. Genetic Metrics in High Transmission Settings: *“...diversity indices...seem to get saturated...”*.

While higher transmission settings may be saturated for some genetic metrics, disentangling polygenomic infections to reveal haplotypes and other metrics like co-transmission or superinfection may be informative in high burden settings.

Reviewer #3:

1. Study Goals: *“...unclear how health policy can be based on such limited data...”*

The intention of this study was to provide detailed baseline pan-Senegal genetic metrics that can be used for comparison in subsequent years in alignment with interventions intended to reduce the malaria burden in Senegal. These data are generated in collaboration with the Senegal NMCP and used for their internal operational planning and evaluation.

2. Vaccine Efficacy: *“...genomic information to predict efficacy of malaria vaccines...”*

There are many applications for genomic surveillance including emerging threats like drug resistance that inform these operational activities (planning and consequences). This study was not designed to address vaccine efficacy but used genetic data to address the emerging threat of drug resistance.

REVIEWERS' COMMENTS

Reviewer #1 (Remarks to the Author):

I thank the authors for their detailed responses, I particularly appreciated the discussion on migration around figure 2B (incidentally, there is a missing word in the sentence beginning "In addition, many of our study sites..."). I find all the issues raised with the initial submission have been addressed

Reviewer #2 (Remarks to the Author):

All my queries have been sufficiently addressed